# Embed to Control: A Locally Linear Latent Dynamics Model for Control from Raw Images

**Manuel Watter**[*]     **Jost Tobias Springenberg**[*]     **Martin Riedmiller**
**Joschka Boedecker**                                        Google DeepMind
University of Freiburg, Germany                               London, UK
{watterm,springj,jboedeck}@cs.uni-freiburg.de     riedmiller@google.com

## Abstract

We introduce Embed to Control (E2C), a method for model learning and control of non-linear dynamical systems from raw pixel images. E2C consists of a deep generative model, belonging to the family of variational autoencoders, that learns to generate image trajectories from a latent space in which the dynamics is constrained to be locally linear. Our model is derived directly from an optimal control formulation in latent space, supports long-term prediction of image sequences and exhibits strong performance on a variety of complex control problems.

## 1   Introduction

Control of non-linear dynamical systems with continuous state and action spaces is one of the key problems in robotics and, in a broader context, in reinforcement learning for autonomous agents. A prominent class of algorithms that aim to solve this problem are model-based locally optimal (stochastic) control algorithms such as iLQG control [1, 2], which approximate the general non-linear control problem via local linearization. When combined with receding horizon control [3], and machine learning methods for learning approximate system models, such algorithms are powerful tools for solving complicated control problems [3, 4, 5]; however, they either rely on a known system model or require the design of relatively low-dimensional state representations. For real *autonomous* agents to succeed, we ultimately need algorithms that are capable of controlling complex dynamical systems from *raw sensory input (e.g. images) only*. In this paper we tackle this difficult problem.

If stochastic optimal control (SOC) methods were applied directly to control from raw image data, they would face two major obstacles. First, sensory data is usually high-dimensional – i.e. images with thousands of pixels – rendering a naive SOC solution computationally infeasible. Second, the image content is typically a highly non-linear function of the system dynamics underlying the observations; thus model identification and control of this dynamics are non-trivial.

While both problems could, in principle, be addressed by designing more advanced SOC algorithms we approach the "optimal control from raw images" problem differently: turning the problem of locally optimal control in high-dimensional non-linear systems into one of identifying a low-dimensional latent state space, in which locally optimal control can be performed robustly and easily. To learn such a latent space we propose a new deep generative model belonging to the class of variational autoencoders [6, 7] that is derived from an iLQG formulation in latent space. The resulting *Embed to Control (E2C)* system is a probabilistic generative model that holds a belief over viable trajectories in sensory space, allows for accurate long-term planning in latent space, and is trained fully unsupervised. We demonstrate the success of our approach on four challenging tasks for control from raw images and compare it to a range of methods for unsupervised representation learning. As an aside, we also validate that deep up-convolutional networks [8, 9] are powerful generative models for large images.

---

[*]Authors contributed equally.

## 2 The Embed to Control (E2C) model

We briefly review the problem of SOC for dynamical systems, introduce approximate locally optimal control in latent space, and finish with the derivation of our model.

### 2.1 Problem Formulation

We consider the control of unknown dynamical systems of the form

$$\mathbf{s}_{t+1} = f(\mathbf{s}_t, \mathbf{u}_t) + \boldsymbol{\xi}, \ \ \boldsymbol{\xi} \sim \mathcal{N}(0, \boldsymbol{\Sigma}_{\boldsymbol{\xi}}), \tag{1}$$

where $t$ denotes the time steps, $\mathbf{s}_t \in \mathbb{R}^{n_s}$ the system state, $\mathbf{u}_t \in \mathbb{R}^{n_u}$ the applied control and $\boldsymbol{\xi}$ the system noise. The function $f(\mathbf{s}_t, \mathbf{u}_t)$ is an arbitrary, smooth, system dynamics. We equivalently refer to Equation (1) using the notation $P(\mathbf{s}_{t+1}|\mathbf{s}_t, \mathbf{u}_t)$, which we assume to be a multivariate normal distribution $\mathcal{N}(f(\mathbf{s}_t, \mathbf{u}_t), \boldsymbol{\Sigma}_\xi)$. We further assume that we are only given access to visual depictions $\mathbf{x}_t \in \mathbb{R}^{n_x}$ of state $\mathbf{s}_t$. This restriction requires solving a joint state identification and control problem. For simplicity we will in the following assume that $\mathbf{x}_t$ is a fully observed depiction of $\mathbf{s}_t$, but relax this assumption later.

Our goal then is to infer a low-dimensional latent state space model in which optimal control can be performed. That is, we seek to learn a function $m$, mapping from high-dimensional images $\mathbf{x}_t$ to low-dimensional vectors $\mathbf{z}_t \in \mathbb{R}^{n_z}$ with $n_z \ll n_x$, such that the control problem can be solved using $\mathbf{z}_t$ instead of $\mathbf{x}_t$:

$$\mathbf{z}_t = m(\mathbf{x}_t) + \boldsymbol{\omega}, \ \ \boldsymbol{\omega} \sim \mathcal{N}(0, \boldsymbol{\Sigma}_{\boldsymbol{\omega}}), \tag{2}$$

where $\boldsymbol{\omega}$ accounts for system noise; or equivalently $\mathbf{z}_t \sim \mathcal{N}(m(\mathbf{x}_t), \boldsymbol{\Sigma}_{\boldsymbol{\omega}})$. Assuming for the moment that such a function can be learned (or approximated), we will first define SOC in a latent space and introduce our model thereafter.

### 2.2 Stochastic locally optimal control in latent spaces

Let $\mathbf{z}_t \in \mathbb{R}^{n_z}$ be the inferred latent state from image $\mathbf{x}_t$ of state $\mathbf{s}_t$ and $f^{\mathrm{lat}}(\mathbf{z}_t, \mathbf{u}_t)$ the transition dynamics in latent space, i.e., $\mathbf{z}_{t+1} = f^{\mathrm{lat}}(\mathbf{z}_t, \mathbf{u}_t)$. Thus $f^{\mathrm{lat}}$ models the changes that occur in $\mathbf{z}_t$ when control $\mathbf{u}_t$ is applied to the underlying system as a latent space analogue to $f(\mathbf{s}_t, \mathbf{u}_t)$. Assuming $f^{\mathrm{lat}}$ is known, optimal controls for a trajectory of length $T$ in the dynamical system can be derived by minimizing the function $J(\mathbf{z}_{1:T}, \mathbf{u}_{1:T})$ which gives the expected future costs when following $(\mathbf{z}_{1:T}, \mathbf{u}_{1:T})$:

$$J(\mathbf{z}_{1:T}, \mathbf{u}_{1:T}) = \mathbb{E}_{\mathbf{z}} \left[ c_T(\mathbf{z}_T, \mathbf{u}_T) + \sum_{t_0}^{T-1} c(\mathbf{z}_t, \mathbf{u}_t) \right], \tag{3}$$

where $c(\mathbf{z}_t, \mathbf{u}_t)$ are instantaneous costs, $c_T(\mathbf{z}_T, \mathbf{u}_T)$ denotes terminal costs and $\mathbf{z}_{1:T} = \{\mathbf{z}_1, \ldots, \mathbf{z}_T\}$ and $\mathbf{u}_{1:T} = \{\mathbf{u}_1, \ldots, \mathbf{u}_T\}$ are state and action sequences respectively. If $\mathbf{z}_t$ contains sufficient information about $\mathbf{s}_t$, i.e., $\mathbf{s}_t$ can be inferred from $\mathbf{z}_t$ alone, and $f^{\mathrm{lat}}$ is differentiable, the cost-minimizing controls can be computed from $J(\mathbf{z}_{1:T}, \mathbf{u}_{1:T})$ via SOC algorithms [10]. These optimal control algorithms approximate the global non-linear dynamics with locally linear dynamics at each time step $t$. Locally optimal actions can then be found in closed form. Formally, given a reference trajectory $\bar{\mathbf{z}}_{1:T}$ – the current estimate for the optimal trajectory – together with corresponding controls $\bar{\mathbf{u}}_{1:T}$ the system is linearized as

$$\mathbf{z}_{t+1} = \mathbf{A}(\bar{\mathbf{z}}_t)\mathbf{z}_t + \mathbf{B}(\bar{\mathbf{z}}_t)\mathbf{u}_{t+1} + \mathbf{o}(\bar{\mathbf{z}}_t) + \boldsymbol{\omega}, \ \ \boldsymbol{\omega} \sim \mathcal{N}(0, \boldsymbol{\Sigma}_{\boldsymbol{\omega}}), \tag{4}$$

where $\mathbf{A}(\bar{\mathbf{z}}_t) = \frac{\delta f^{\mathrm{lat}}(\bar{\mathbf{z}}_t, \bar{\mathbf{u}}_t)}{\delta \bar{\mathbf{z}}_t}$, $\mathbf{B}(\bar{\mathbf{z}}_t) = \frac{\delta f^{\mathrm{lat}}(\bar{\mathbf{z}}_t, \bar{\mathbf{u}}_t)}{\delta \bar{\mathbf{u}}_t}$ are local Jacobians, and $\mathbf{o}(\bar{\mathbf{z}}_t)$ is an offset. To enable efficient computation of the local controls we assume the costs to be a quadratic function of the latent representation

$$c(\mathbf{z}_t, \mathbf{u}_t) = (\mathbf{z}_t - \mathbf{z}_{\mathrm{goal}})^T \mathbf{R}_z (\mathbf{z}_t - \mathbf{z}_{\mathrm{goal}}) + \mathbf{u}_t^T \mathbf{R}_u \mathbf{u}_t, \tag{5}$$

where $\mathbf{R}_z \in \mathbb{R}^{n_z \times n_z}$ and $\mathbf{R}_u \in \mathbb{R}^{n_u \times n_u}$ are cost weighting matrices and $\mathbf{z}_{\mathrm{goal}}$ is the inferred representation of the goal state. We also assume $c_T(\mathbf{z}_T, \mathbf{u}_T) = c(\mathbf{z}_T, \mathbf{u}_T)$ throughout this paper. In combination with Equation (4) this gives us a local *linear-quadratic-Gaussian* formulation at each time step $t$ which can be solved by SOC algorithms such as iterative linear-quadratic regulation (iLQR) [11] or approximate inference control (AICO) [12]. The result of this trajectory optimization step is a locally optimal trajectory with corresponding control sequence $(\mathbf{z}_{1:T}^*, \mathbf{u}_{1:T}^*) \approx \arg\min_{\substack{\mathbf{z}_{1:T} \\ \mathbf{u}_{1:T}}} J(\mathbf{z}_{1:T}, \mathbf{u}_{1:T})$.

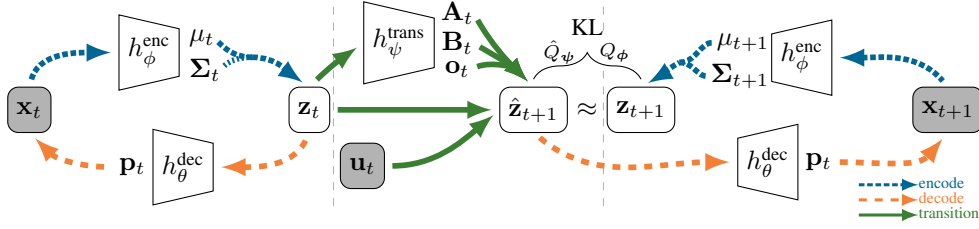

Figure 1: The information flow in the E2C model. From left to right, we encode and decode an image $\mathbf{x}_t$ with the networks $h_{\phi}^{\text{enc}}$ and $h_{\theta}^{\text{dec}}$, where we use the latent code $\mathbf{z}_t$ for the transition step. The $h_{\psi}^{\text{trans}}$ network computes the local matrices $\mathbf{A}_t, \mathbf{B}_t, \mathbf{o}_t$ with which we can predict $\hat{\mathbf{z}}_{t+1}$ from $\mathbf{z}_t$ and $\mathbf{u}_t$. Similarity to the encoding $\mathbf{z}_{t+1}$ is enforced by a KL divergence on their distributions and reconstruction is again performed by $h_{\theta}^{\text{dec}}$.

## 2.3 A locally linear latent state space model for dynamical systems

Starting from the SOC formulation, we now turn to the problem of learning an appropriate low-dimensional latent representation $\mathbf{z}_t \sim P(Z_t|m(\mathbf{x}_t), \boldsymbol{\Sigma}_{\boldsymbol{\omega}})$ of $\mathbf{x}_t$. The representation $\mathbf{z}_t$ has to fulfill three properties: (i) it must capture sufficient information about $\mathbf{x}_t$ (enough to enable reconstruction); (ii) it must allow for accurate prediction of the next latent state $\mathbf{z}_{t+1}$ and thus, implicitly, of the next observation $\mathbf{x}_{t+1}$; (iii) the prediction $f^{\text{lat}}$ of the next latent state must be locally linearizable *for all valid control magnitudes* $\mathbf{u}_t$. Given some representation $\mathbf{z}_t$, properties (ii) and (iii) in particular require us to capture possibly highly non-linear changes of the latent representation due to transformations of the observed scene induced by control commands. Crucially, these are particularly hard to model and subsequently linearize. We circumvent this problem by taking a more direct approach: instead of learning a latent space $\mathbf{z}$ and transition model $f^{\text{lat}}$ which are then linearized and combined with SOC algorithms, we directly impose desired transformation properties on the representation $\mathbf{z}_t$ during learning. We will select these properties such that prediction in the latent space as well as locally linear inference of the next observation according to Equation (4) are easy.

The transformation properties that we desire from a latent representation can be formalized directly from the iLQG formulation given in Section 2.2 . Formally, following Equation (2), let the latent representation be Gaussian $P(Z|X) = \mathcal{N}(m(\mathbf{x}_t), \boldsymbol{\Sigma}_{\boldsymbol{\omega}})$. To infer $\mathbf{z}_t$ from $\mathbf{x}_t$ we first require a method for sampling latent states. Ideally, we would generate samples directly from the unknown true posterior $P(Z|X)$, which we, however, have no access to. Following the variational Bayes approach (see Jordan et al. [13] for an overview) we resort to sampling $\mathbf{z}_t$ from an approximate posterior distribution $Q_{\phi}(Z|X)$ with parameters $\phi$.

**Inference model for $Q_{\phi}$.** In our work this is always a diagonal Gaussian distribution $Q_{\phi}(Z|X) = \mathcal{N}(\boldsymbol{\mu}_t, \text{diag}(\boldsymbol{\sigma}_t^2))$, whose mean $\boldsymbol{\mu}_t \in \mathbb{R}^{n_z}$ and covariance $\boldsymbol{\Sigma}_t = \text{diag}(\boldsymbol{\sigma}_t^2) \in \mathbb{R}^{n_z \times n_z}$ are computed by an encoding neural network with outputs

$$\boldsymbol{\mu}_t = \mathbf{W}_{\boldsymbol{\mu}} h_{\phi}^{\text{enc}}(\mathbf{x}_t) + \mathbf{b}_{\boldsymbol{\mu}}, \tag{6}$$

$$\log \boldsymbol{\sigma}_t = \mathbf{W}_{\boldsymbol{\sigma}} h_{\phi}^{\text{enc}}(\mathbf{x}_t) + \mathbf{b}_{\boldsymbol{\sigma}}, \tag{7}$$

where $h_{\phi}^{\text{enc}} \in \mathbb{R}^{n_e}$ is the activation of the last hidden layer and where $\phi$ is given by the set of all learnable parameters of the encoding network, including the weight matrices $\mathbf{W}_{\boldsymbol{\mu}}, \mathbf{W}_{\boldsymbol{\sigma}}$ and biases $\mathbf{b}_{\boldsymbol{\mu}}, \mathbf{b}_{\boldsymbol{\sigma}}$. Parameterizing the mean and variance of a Gaussian distribution based on a neural network gives us a natural and very expressive model for our latent space. It additionally comes with the benefit that we can use the *reparameterization trick* [6, 7] to backpropagate gradients of a loss function based on samples through the latent distribution.

**Generative model for $P_{\theta}$.** Using the approximate posterior distribution $Q_{\phi}$ we generate observed samples (images) $\tilde{\mathbf{x}}_t$ and $\tilde{\mathbf{x}}_{t+1}$ from latent samples $\mathbf{z}_t$ and $\mathbf{z}_{t+1}$ by enforcing a locally linear relationship in latent space according to Equation (4), yielding the following generative model

$$\begin{aligned} \mathbf{z}_t &\sim Q_{\phi}(Z \mid X) &&= \mathcal{N}(\boldsymbol{\mu}_t, \boldsymbol{\Sigma}_t), \\ \hat{\mathbf{z}}_{t+1} &\sim \hat{Q}_{\psi}(\hat{Z} \mid Z, \mathbf{u}) &&= \mathcal{N}(\mathbf{A}_t \boldsymbol{\mu}_t + \mathbf{B}_t \mathbf{u}_t + \mathbf{o}_t, \mathbf{C}_t), \\ \tilde{\mathbf{x}}_t, \tilde{\mathbf{x}}_{t+1} &\sim P_{\theta}(X \mid Z) &&= Bernoulli(\mathbf{p}_t), \end{aligned} \tag{8}$$

where $\hat{Q}_{\psi}$ is the *next latent state* posterior distribution, which exactly follows the linear form required for stochastic optimal control. With $\boldsymbol{\omega}_t \sim \mathcal{N}(\mathbf{0}, \mathbf{H}_t)$ as an estimate of the system noise,

$\mathbf{C}$ can be decomposed as $\mathbf{C}_t = \mathbf{A}_t \mathbf{\Sigma}_t \mathbf{A}_t^T + \mathbf{H}_t$. Note that while the transition dynamics in our generative model operates on the inferred latent space, it takes untransformed controls into account. That is, we aim to learn a latent space such that the transition dynamics in $\mathbf{z}$ linearizes the non-linear observed dynamics in $\mathbf{x}$ and is locally linear in the applied controls $\mathbf{u}$. Reconstruction of an image from $\mathbf{z}_t$ is performed by passing the sample through multiple hidden layers of a decoding neural network which computes the mean $\mathbf{p}_t$ of the generative Bernoulli distribution[1] $P_{\boldsymbol{\theta}}(X|Z)$ as

$$\mathbf{p}_t = \mathbf{W}_{\mathbf{p}} h_{\boldsymbol{\theta}}^{\text{dec}}(\mathbf{z}_t) + \mathbf{b}_{\mathbf{p}}, \tag{9}$$

where $h_{\boldsymbol{\theta}}^{\text{dec}}(\mathbf{z}_t) \in \mathbb{R}^{n_d}$ is the response of the last hidden layer in the decoding network. The set of parameters for the decoding network, including weight matrix $\mathbf{W}_{\mathbf{p}}$ and bias $\mathbf{b}_{\mathbf{p}}$, then make up the learned generative parameters $\boldsymbol{\theta}$.

**Transition model for $\hat{Q}_{\psi}$.** What remains is to specify how the linearization matrices $\mathbf{A}_t \in \mathbb{R}^{n_z \times n_z}$, $\mathbf{B}_t \in \mathbb{R}^{n_z \times n_u}$ and offset $\mathbf{o}_t \in \mathbb{R}^{n_z}$ are predicted. Following the same approach as for distribution means and covariance matrices, we predict all local transformation parameters from samples $\mathbf{z}_t$ based on the hidden representation $h_{\psi}^{\text{trans}}(\mathbf{z}_t) \in \mathbb{R}^{n_t}$ of a third neural network with parameters $\psi$ – to which we refer as the transformation network. Specifically, we parametrize the transformation matrices and offset as

$$\begin{aligned}
\text{vec}[\mathbf{A}_t] &= \mathbf{W}_A \, h_{\psi}^{\text{trans}}(\mathbf{z}_t) + \mathbf{b}_A, \\
\text{vec}[\mathbf{B}_t] &= \mathbf{W}_B \, h_{\psi}^{\text{trans}}(\mathbf{z}_t) + \mathbf{b}_B, \\
\mathbf{o}_t &= \mathbf{W}_o \, h_{\psi}^{\text{trans}}(\mathbf{z}_t) + \mathbf{b}_o,
\end{aligned} \tag{10}$$

where vec denotes vectorization and therefore $\text{vec}[\mathbf{A}_t] \in \mathbb{R}^{(n_z^2)}$ and $\text{vec}[\mathbf{B}_t] \in \mathbb{R}^{(n_z \cdot n_u)}$. To circumvent estimating the full matrix $\mathbf{A}_t$ of size $n_z \times n_z$, we can choose it to be a perturbation of the identity matrix $\mathbf{A}_t = (\mathbf{I} + \mathbf{v}_t \mathbf{r}_t^T)$ which reduces the parameters to be estimated for $\mathbf{A}_t$ to $2n_z$.

A sketch of the complete architecture is shown in Figure 1. It also visualizes an additional constraint that is essential for learning a representation for long-term predictions: we require samples $\hat{\mathbf{z}}_{t+1}$ from the state transition distribution $\hat{Q}_{\psi}$ to be similar to the encoding of $\mathbf{x}_{t+1}$ through $Q_{\phi}$. While it might seem that just learning a perfect reconstruction of $\mathbf{x}_{t+1}$ from $\hat{\mathbf{z}}_{t+1}$ is enough, we require multi-step predictions for planning in $Z$ which *must correspond* to valid trajectories in the observed space $X$. Without enforcing similarity between samples from $\hat{Q}_{\psi}$ and $Q_{\phi}$, following a transition in latent space from $\mathbf{z}_t$ with action $\mathbf{u}_t$ may lead to a point $\hat{\mathbf{z}}_{t+1}$, from which reconstruction of $\mathbf{x}_{t+1}$ is possible, but that is not a valid encoding (i.e. the model will never encode any image as $\hat{\mathbf{z}}_{t+1}$). Executing another action in $\hat{\mathbf{z}}_{t+1}$ then does not result in a valid latent state – since the transition model is conditional on samples coming from the inference network – and thus long-term predictions fail. In a nutshell, such a divergence between encodings and the transition model results in a generative model that does not accurately model the Markov chain formed by the observations.

## 2.4 Learning via stochastic gradient variational Bayes

For training the model we use a data set $\mathcal{D} = \{(\mathbf{x}_1, \mathbf{u}_1, \mathbf{x}_2), \ldots, (\mathbf{x}_{T-1}, \mathbf{u}_{T-1}, \mathbf{x}_T)\}$ containing observation tuples with corresponding controls obtained from interactions with the dynamical system. Using this data set, we learn the parameters of the inference, transition and generative model by minimizing a variational bound on the true data negative log-likelihood $-\log P(\mathbf{x}_t, \mathbf{u}_t, \mathbf{x}_{t+1})$ plus an additional constraint on the latent representation. The complete loss function[2] is given as

$$\mathcal{L}(\mathcal{D}) = \sum_{(\mathbf{x}_t, \mathbf{u}_t, \mathbf{x}_{t+1}) \in \mathcal{D}} \mathcal{L}^{\text{bound}}(\mathbf{x}_t, \mathbf{u}_t, \mathbf{x}_{t+1}) + \lambda \, \text{KL}\left(\hat{Q}_{\psi}(\hat{Z} \mid \boldsymbol{\mu}_t, \mathbf{u}_t) \big\| Q_{\phi}(Z \mid \mathbf{x}_{t+1})\right). \tag{11}$$

The first part of this loss is the per-example variational bound on the log-likelihood

$$\mathcal{L}^{\text{bound}}(\mathbf{x}_t, \mathbf{u}_t, \mathbf{x}_{t+1}) = \mathbb{E}_{\substack{\mathbf{z}_t \sim Q_{\phi} \\ \hat{\mathbf{z}}_{t+1} \sim \hat{Q}_{\psi}}} \left[-\log P_{\boldsymbol{\theta}}(\mathbf{x}_t|\mathbf{z}_t) - \log P_{\boldsymbol{\theta}}(\mathbf{x}_{t+1}|\hat{\mathbf{z}}_{t+1})\right] + \text{KL}(Q_{\phi}\|P(Z)), \tag{12}$$

where $Q_{\phi}$, $P_{\boldsymbol{\theta}}$ and $\hat{Q}_{\psi}$ are the parametric inference, generative and transition distributions from Section 2.3 and $P(Z_t)$ is a prior on the approximate posterior $Q_{\phi}$; which we always chose to be

an isotropic Gaussian distribution with mean zero and unit variance. The second KL divergence in Equation (11) is an additional contraction term with weight $\lambda$, that enforces agreement between the transition and inference models. This term is essential for establishing a Markov chain in latent space that corresponds to the real system dynamics (see Section 2.3 above for an in depth discussion). This KL divergence can also be seen as a prior on the latent transition model. Note that all KL terms can be computed analytically for our model (see supplementary for details).

During training we approximate the expectation in $\mathcal{L}(\mathcal{D})$ via sampling. Specifically, we take one sample $\mathbf{z}_t$ for each input $\mathbf{x}_t$ and transform that sample using Equation (10) to give a valid sample $\hat{\mathbf{z}}_{t+1}$ from $\hat{Q}_\psi$. We then jointly learn all parameters of our model by minimizing $\mathcal{L}(\mathcal{D})$ using SGD.

## 3 Experimental Results

We evaluate our model on four visual tasks: an agent in a plane with obstacles, a visual version of the classic inverted pendulum swing-up task, balancing a cart-pole system, and control of a three-link arm with larger images. These are described in detail below.

### 3.1 Experimental Setup

**Model training.** We consider two different network types for our model: Standard fully connected neural networks with up to three layers, which work well for moderately sized images, are used for the planar and swing-up experiments; A deep convolutional network for the encoder in combination with an up-convolutional network as the decoder which, in accordance with recent findings from the literature [8, 9], we found to be an adequate model for larger images. Training was performed using Adam [14] throughout all experiments. The training data set $\mathcal{D}$ for all tasks was generated by randomly sampling $N$ state observations and actions with corresponding successor states. For the plane we used $N = 3,000$ samples, for the inverted pendulum and cart-pole system we used $N = 15,000$ and for the arm $N = 30,000$. A complete list of architecture parameters and hyperparameter choices as well as an in-depth explanation of the up-convolutional network are specified in the supplementary material. We will make our code and a video containing controlled trajectories for all systems available under http://ml.informatik.uni-freiburg.de/research/e2c.

**Model variants.** In addition to the Embed to Control (E2C) dynamics model derived above, we also consider two variants: By removing the latent dynamics network $h_\psi^{\text{trans}}$, i.e. setting its output to one in Equation (10) – we obtain a variant in which $\mathbf{A}_t$, $\mathbf{B}_t$ and $\mathbf{o}_t$ are estimated as globally linear matrices (Global E2C). If we instead replace the transition model with a network estimating the dynamics as a non-linear function $\hat{f}^{\text{lat}}$ and only linearize during planning, estimating $\mathbf{A}_t$, $\mathbf{B}_t$, $\mathbf{o}_t$ as Jacobians to $\hat{f}^{\text{lat}}$ as described in Section 2.2, we obtain a variant with nonlinear latent dynamics.

**Baseline models.** For a thorough comparison and to exhibit the complicated nature of the tasks, we also test a set of baseline models on the plane and the inverted pendulum task (using the same architecture as the E2C model): a standard variational autoencoder (VAE) and a deep autoencoder (AE) are trained on the autoencoding subtask for visual problems. That is, given a data set $\mathcal{D}$ used for training our model, we remove all actions from the tuples in $\mathcal{D}$ and disregard temporal context between images. After autoencoder training we learn a dynamics model in latent space, approximating $f^{\text{lat}}$ from Section 2.2. We also consider a VAE variant with a slowness term on the latent representation – a full description of this variant is given in the supplementary material.

**Optimal control algorithms.** To perform optimal control in the latent space of different models, we employ two trajectory optimization algorithms: iterative linear quadratic regulation (iLQR) [11] (for the plane and inverted pendulum) and approximate inference control (AICO) [12] (all other experiments). For all VAEs both methods operate on the mean of distributions $Q_\phi$ and $\hat{Q}_\psi$. AICO additionally makes use of the local Gaussian covariances $\mathbf{\Sigma}_t$ and $\mathbf{C}_t$. Except for the experiments on the planar system, control was performed in a model predictive control fashion using the receding horizon scheme introduced in [3]. To obtain closed loop control given an image $\mathbf{x}_t$, it is first passed through the encoder to obtain the latent state $\mathbf{z}_t$. A locally optimal trajectory is subsequently found by optimizing $(\mathbf{z}_{t:t+T}^*, \mathbf{u}_{t:t+T}^*) \approx \arg\min_{\substack{\mathbf{z}_{t:t+T} \\ \mathbf{u}_{t:t+T}}} J(\mathbf{z}_{t:t+T}, \mathbf{u}_{t:t+T})$ with fixed, small horizon $T$ (with $T = 10$ unless noted otherwise). Controls $\mathbf{u}_t^*$ are applied to the system and a transition to $\mathbf{z}_{t+1}$ is observed (by encoding the next image $\mathbf{x}_{t+1}$). Then a new control sequence – with horizon

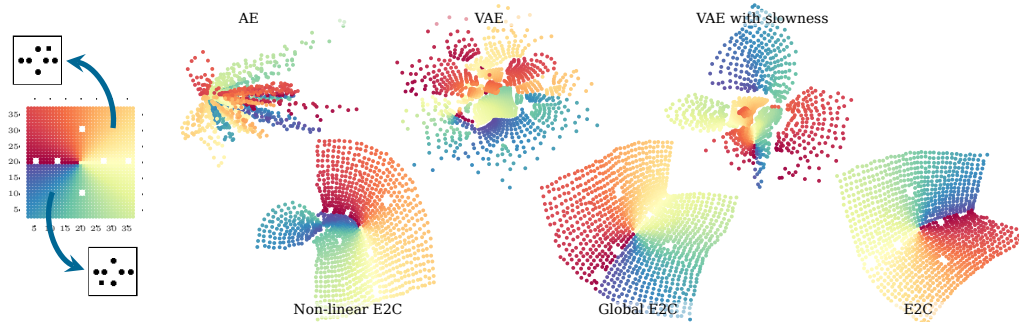

Figure 2: The true state space of the planar system (left) with examples (obstacles encoded as circles) and the inferred spaces (right) of different models. The spaces are spanned by generating images for every valid position of the agent and embedding them with the respective encoders.

$T$ – starting in $\mathbf{z}_{t+1}$ is found using the last estimated trajectory as a bootstrap. Note that planning is performed entirely in the latent state *without access to any observations* except for the depiction of the current state. To compute the cost function $c(\mathbf{z}_t, \mathbf{u}_t)$ required for trajectory optimization in $\mathbf{z}$ we assume knowledge of the observation $\mathbf{x}_{\text{goal}}$ of the goal state $\mathbf{s}_{\text{goal}}$. This observation is then transformed into latent space and costs are computed according to Equation (5).

## 3.2 Control in a planar system

The agent in the planar system can move in a bounded two-dimensional plane by choosing a continuous offset in x- and y-direction. The high-dimensional representation of a state is a $40 \times 40$ black-and-white image. Obstructed by six circular obstacles, the task is to move to the bottom right of the image, starting from a random x position at the top of the image. The encodings of obstacles are obtained prior to planning and an additional quadratic cost term is penalizing proximity to them.

A depiction of the observations on which control is performed – together with their corresponding state values and embeddings into latent space – is shown in Figure 2. The figure also clearly shows a fundamental advantage the E2C model has over its competitors: While the separately trained autoencoders make for aesthetically pleasing pictures, the models failed to discover the underlying structure of the state space, complicating dynamics estimation and largely invalidating costs based on distances in said space. Including the latent dynamics constraints in these end-to-end models on the other hand, yields latent spaces approaching the optimal planar embedding.

We test the long-term accuracy by accumulating latent and real trajectory costs to quantify whether the imagined trajectory reflects reality. The results for all models when starting from random positions at the top and executing 40 pre-computed actions are summarized in Table 1 – using a seperate test set for evaluating reconstructions. While all methods achieve a low reconstruction loss, the difference in accumulated real costs per trajectory show the superiority of the E2C model. Using the globally or locally linear E2C model, trajectories planned in latent space are as good as trajectories planned on the real state. All models besides E2C fail to give long-term predictions that result in good performance.

## 3.3 Learning swing-up for an inverted pendulum

We next turn to the task of controlling the classical inverted pendulum system [15] from images. We create depictions of the state by rendering a fixed length line starting from the center of the image at an angle corresponding to the pendulum position. The goal in this task is to swing-up and balance an underactuated pendulum from a resting position (pendulum hanging down). Exemplary observations and reconstructions for this system are given in Figure 3(d). In the visual inverted pendulum task our algorithm faces two additional difficulties: the observed space is non-Markov, as the angular velocity cannot be inferred from a single image, and second, discretization errors due to rendering pendulum angles as small 48x48 pixel images make exact control difficult. To restore the Markov property, we stack two images (as input channels), thus observing a one-step history.

Figure 3 shows the topology of the latent space for our model, as well as one sample trajectory in true state and latent space. The fact that the model can learn a meaningful embedding, separating

Table 1: Comparison between different approaches to model learning from raw pixels for the planar and pendulum system. We compare all models with respect to their prediction quality on a test set of sampled transitions and with respect to their performance when combined with SOC (trajectory cost for control from different start states). Note that trajectory costs in latent space are not necessarily comparable. The "real" trajectory cost was computed on the dynamics of the simulator while executing planned actions. For the true models for $\mathbf{s}_t$, real trajectory costs were $20.24 \pm 4.15$ for the planar system, and $9.8 \pm 2.4$ for the pendulum. Success was defined as reaching the goal state and staying $\epsilon$-close to it for the rest of the trajectory (if non terminating). All statistics quantify over 5/30 (plane/pendulum) different starting positions. A † marks separately trained dynamics networks.

| Algorithm | State Loss $\log \mathbf{p}(\mathbf{x_t}\vert\mathbf{\hat{x}_t})$ | Next State Loss $\log \mathbf{p}(\mathbf{x_{t+1}}\vert\mathbf{\hat{x}_t}, \mathbf{u_t})$ | Trajectory Cost Latent | Real | Success percent |
|---|---|---|---|---|---|
| **Planar System** | | | | | |
| AE† | $11.5 \pm 97.8$ | $3538.9 \pm 1395.2$ | $1325.6 \pm 81.2$ | $273.3 \pm 16.4$ | $0\,\%$ |
| VAE† | $3.6 \pm 18.9$ | $652.1 \pm 930.6$ | $43.1 \pm 20.8$ | $91.3 \pm 16.4$ | $0\,\%$ |
| VAE + slowness† | $10.5 \pm 22.8$ | $104.3 \pm 235.8$ | $47.1 \pm 20.5$ | $89.1 \pm 16.4$ | $0\,\%$ |
| Non-linear E2C | $8.3 \pm 5.5$ | $11.3 \pm 10.1$ | $19.8 \pm 9.8$ | $42.3 \pm 16.4$ | $96.6\,\%$ |
| Global E2C | $\mathbf{6.9 \pm 3.2}$ | $\mathbf{9.3 \pm 4.6}$ | $12.5 \pm 3.9$ | $27.3 \pm 9.7$ | $\mathbf{100}\,\%$ |
| **E2C** | $7.7 \pm 2.0$ | $9.7 \pm 3.2$ | $10.3 \pm 2.8$ | $\mathbf{25.1 \pm 5.3}$ | $\mathbf{100}\,\%$ |
| **Inverted Pendulum Swing-Up** | | | | | |
| AE† | $8.9 \pm 100.3$ | $13433.8 \pm 6238.8$ | $1285.9 \pm 355.8$ | $194.7 \pm 44.8$ | $0\,\%$ |
| VAE† | $7.5 \pm 47.7$ | $8791.2 \pm 17356.9$ | $497.8 \pm 129.4$ | $237.2 \pm 41.2$ | $0\,\%$ |
| VAE + slowness† | $26.5 \pm 18.0$ | $779.7 \pm 633.3$ | $419.5 \pm 85.8$ | $188.2 \pm 43.6$ | $0\,\%$ |
| E2C no latent KL | $64.4 \pm 32.8$ | $87.7 \pm 64.2$ | $489.1 \pm 87.5$ | $213.2 \pm 84.3$ | $0\,\%$ |
| Non-linear E2C | $\mathbf{59.6 \pm 25.2}$ | $\mathbf{72.6 \pm 34.5}$ | $313.3 \pm 65.7$ | $37.4 \pm 12.4$ | $63.33\,\%$ |
| Global E2C | $115.5 \pm 56.9$ | $125.3 \pm 62.6$ | $628.1 \pm 45.9$ | $125.1 \pm 10.7$ | $0\,\%$ |
| **E2C** | $84.0 \pm 50.8$ | $89.3 \pm 42.9$ | $275.0 \pm 16.6$ | $\mathbf{15.4 \pm 3.4}$ | $\mathbf{90}\,\%$ |

velocities and positions, from this data is remarkable (no other model recovered this shape). Table 1 again compares the different models quantitatively. While the E2C model is not the best in terms of reconstruction performance, it is the only model resulting in stable swing-up *and* balance behavior. We explain the failure of the other models with the fact that the non-linear latent dynamics model cannot be guaranteed to be linearizable for all control magnitudes, resulting in undesired behavior around unstable fixpoints of the real system dynamics, and that for this task a globally linear dynamics model is inadequate.

## 3.4 Balancing a cart-pole and controlling a simulated robot arm

Finally, we consider control of two more complex dynamical systems from images using a six layer convolutional inference and six layer up-convolutional generative network, resulting in a *12-layer deep* path from input to reconstruction. Specifically, we control a visual version of the classical cart-pole system [16] from a history of two $80 \times 80$ pixel images as well as a three-link planar robot arm based on a history of two $128 \times 128$ pixel images. The latent space was set to be 8-dimensional in both experiments. The real state dimensionality for the cart-pole is four and is controlled using one

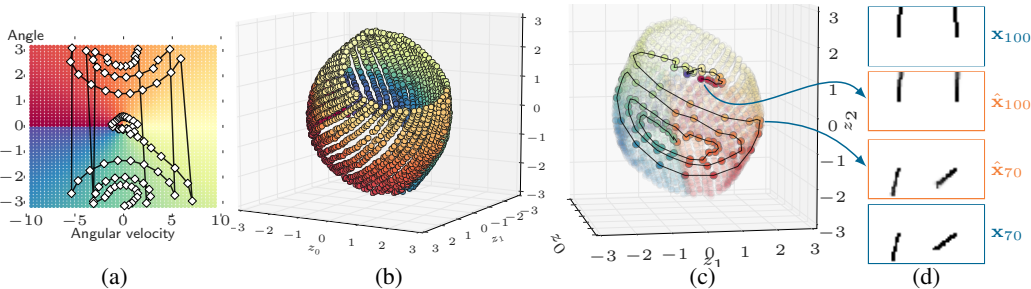

(a)    (b)    (c)    (d)

Figure 3: (a) The true state space of the inverted pendulum task overlaid with a successful trajectory taken by the E2C agent. (b) The learned latent space. (c) The trajectory from (a) traced out in the latent space. (d) Images $\mathbf{x}$ and reconstructions $\hat{\mathbf{x}}$ showing current positions (right) and history (left).

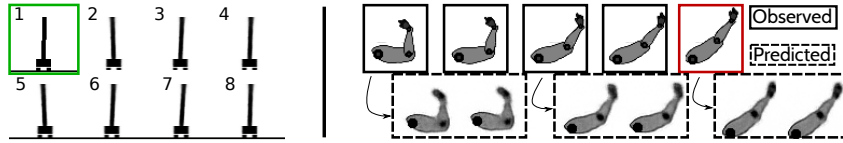

Figure 4: Left: Trajectory from the cart-pole domain. Only the first image (green) is "real", all other images are "dreamed up" by our model. Notice discretization artifacts present in the real image. Right: Exemplary observed (with history image omitted) and predicted images (including the history image) for a trajectory in the visual robot arm domain with the goal marked in red.

action, while for the arm the real state can be described in 6 dimensions (joint angles and velocities) and controlled using a three-dimensional action vector corresponding to motor torques.

As in previous experiments the E2C model seems to have no problem finding a locally linear embedding of images into latent space in which control can be performed. Figure 4 depicts exemplary images – for both problems – from a trajectory executed by our system. The costs for these trajectories (11.13 for the cart-pole, 85.12 for the arm) are only slightly worse than trajectories obtained by AICO operating on the real system dynamics starting from the same start-state (7.28 and 60.74 respectively). The supplementary material contains additional experiments using these domains.

## 4   Comparison to recent work

In the context of representation learning for control (see Böhmer et al. [17] for a review), deep autoencoders (ignoring state transitions) similar to our baseline models have been applied previously, e.g. by Lange and Riedmiller [18]. A more direct route to control based on image streams is taken by recent work on (model free) deep end-to-end Q-learning for Atari games by Mnih et al. [19], as well as kernel based [20] and deep policy learning for robot control [21].

Close to our approach is a recent paper by Wahlström et al. [22], where autoencoders are used to extract a latent representation for control from images, on which a non-linear model of the forward dynamics is learned. Their model is trained jointly and is thus similar to the non-linear E2C variant in our comparison. In contrast to our model, their formulation requires PCA pre-processing and does neither ensure that long-term predictions in latent space do not diverge, nor that they are linearizable.

As stated above, our system belongs to the family of VAEs and is generally similar to recent work such as Kingma and Welling [6], Rezende et al. [7], Gregor et al. [23], Bayer and Osendorfer [24]. Two additional parallels between our work and recent advances for training deep neural networks can be observed. First, the idea of enforcing desired transformations in latent space during learning – such that the data becomes easy to model – has appeared several times already in the literature. This includes the development of transforming auto-encoders [25] and recent probabilistic models for images [26, 27]. Second, learning relations between pairs of images – although *without control* – has received considerable attention from the community during the last years [28, 29]. In a broader context our model is related to work on state estimation in Markov decision processes (see Langford et al. [30] for a discussion) through, e.g., hidden Markov models and Kalman filters [31, 32].

## 5   Conclusion

We presented Embed to Control (E2C), a system for stochastic optimal control on high-dimensional image streams. Key to the approach is the extraction of a latent dynamics model which is constrained to be locally linear in its state transitions. An evaluation on four challenging benchmarks revealed that E2C can find embeddings on which control can be performed with ease, reaching performance close to that achievable by optimal control on the real system model.

**Acknowledgments**

We thank A. Radford, L. Metz, and T. DeWolf for sharing code, as well as A. Dosovitskiy for useful discussions. This work was partly funded by a DFG grant within the priority program "Autonomous learning" (SPP1597) and the BrainLinks-BrainTools Cluster of Excellence (grant number EXC 1086). M. Watter is funded through the State Graduate Funding Program of Baden-Württemberg.

## Footnotes

[1] A Bernoulli distribution for $P_{\boldsymbol{\theta}}$ is a common choice when modeling black-and-white images.

[2] Note that this is the loss for the latent state space model and distinct from the SOC costs.

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
