[Supplementary Material · supplementary_embed_to_control.pdf]

# Supplementary Material for "Embed to Control: A Locally Linear Latent Dynamics Model for Control from Raw Images"

**Manuel Watter**[*]     **Jost Tobias Springenberg**[*]          **Martin Riedmiller**
**Joschka Boedecker**                                            Google DeepMind
University of Freiburg, Germany                                   London, UK
{watterm,springj,jboedeck}@cs.uni-freiburg.de      riedmiller@google.com

## Abstract

Supplementary material for the Embed to Control paper. We provide additional details for the derivation of the E2C model in Section 1 . Section 2 exhaustively details our experimental setup, while Section 3 provides additional evaluations and visualizations of planned trajectories and latent spaces.

## 1   Supplementary to the E2C description

### 1.1   State transition matrix factorization and KL Divergence

As alluded to in the main paper, estimation of the full local state transition matrix $\mathbf{A}_t \in \mathbb{R}^{n_z \times n_z}$ from Equation (8) requires the transition network to predict $n_z \times n_z$ parameters. Using an arbitrary state transition matrix also – inconveniently – requires inversion of said matrix for computing the KL divergence penalty from Equation (11) (through which it is hard to backpropagate). We started our experiments using a full matrix (and only approximating all KL divergence terms), but quickly found that a rank one pertubation of the identity matrix could be used instead without loss of performance in any of our benchmarks. To the contrary, the resulting networks have fewer parameters and are thus easier to train. We here give the derivation of this process and how the KL divergence from Equation (11) can be computed. For the reformulation we represent $\mathbf{A}_t$ as $\mathbf{A}_t = \mathbf{I} + \mathbf{v}_t \mathbf{r}_t^T$, therefore only $\mathbf{v}_t$ and $\mathbf{r}_t$ need to be estimated by the transition network, reducing the number of outputs for $\mathbf{A}_t$ from $n_z^2$ to $2n_z$.

The KL divergence between two multivariate Gaussians is given by

$$\mathrm{KL}(\mathcal{N}_0||\mathcal{N}_1) = \frac{1}{2}\left(\mathrm{Tr}\left(\mathbf{\Sigma}_1^{-1}\mathbf{\Sigma}_0\right) + (\boldsymbol{\mu}_1 - \boldsymbol{\mu}_0)^T\mathbf{\Sigma}_1^{-1}(\boldsymbol{\mu}_1 - \boldsymbol{\mu}_0) - k + \log\left(\frac{\det \mathbf{\Sigma}_1}{\det \mathbf{\Sigma}_0}\right)\right). \quad (1)$$

For a simplified notation, such that $\mathrm{KL}(\mathcal{N}_0||\mathcal{N}_1) = \mathrm{KL}(\hat{Q}||Q)$, let us assume

$$\mathcal{N}_0 = \mathcal{N}(\boldsymbol{\mu}_0, \mathbf{A}\mathbf{\Sigma}_0\mathbf{A}^T) = \mathcal{N}(\boldsymbol{\mu}_t, \mathbf{A}_t\mathbf{\Sigma}_t\mathbf{A}_t^T) = \hat{Q},$$
$$\mathcal{N}_1 = \mathcal{N}(\boldsymbol{\mu}_1, \mathbf{\Sigma}_1) = \mathcal{N}(\boldsymbol{\mu}_{t+1}, \mathbf{\Sigma}_{t+1}) = Q.$$

The main point behind the derivation presented in the following, is to make partial derivatives of the above KL divergence efficiently computable. To this end, we cannot take the trace or the determinant via numerical algorithms, because we have to be able to take the gradients in symbolic form. Aside from that, we like to process a batch of samples, so the computation should have a convenient form

---

[*]Authors contributed equally.

and not require excessive amounts of tensor products in between. We start our simplification with the trace term which results in

$$
\begin{aligned}
\operatorname{Tr}\left(\mathbf{\Sigma}_1^{-1}\mathbf{\Sigma}_0\right) &= \operatorname{Tr}\left(\mathbf{\Sigma}_1^{-1}\mathbf{A}\mathbf{\Sigma}_0\mathbf{A}^T\right) \\
&= \operatorname{Tr}\left(\mathbf{\Sigma}_1^{-1}(\mathbf{I}+\mathbf{v}\mathbf{r}^T)\mathbf{\Sigma}_0(\mathbf{I}+\mathbf{v}\mathbf{r}^T)^T\right) \\
&= \operatorname{Tr}\left(\left(\mathbf{\Sigma}_1^{-1}+\mathbf{\Sigma}_1^{-1}\mathbf{v}\mathbf{r}^T\right)\left(\mathbf{\Sigma}_0+\mathbf{\Sigma}_0(\mathbf{v}\mathbf{r}^T)^T\right)\right) \\
&= \operatorname{Tr}\left(\mathbf{\Sigma}_1^{-1}\mathbf{\Sigma}_0+\mathbf{\Sigma}_1^{-1}\mathbf{\Sigma}_0(\mathbf{v}\mathbf{r}^T)^T+\mathbf{\Sigma}_1^{-1}\mathbf{v}\mathbf{r}^T\mathbf{\Sigma}_0+\mathbf{\Sigma}_1^{-1}\mathbf{v}\mathbf{r}^T\mathbf{\Sigma}_0(\mathbf{v}\mathbf{r}^T)^T\right) \\
&\hspace{7cm}\scriptstyle{\operatorname{Tr}(A+B)=\operatorname{Tr}(A)+\operatorname{Tr}(B)} \\
&= \operatorname{Tr}\left(\mathbf{\Sigma}_1^{-1}\mathbf{\Sigma}_0\right)+\operatorname{Tr}\left(\mathbf{\Sigma}_1^{-1}\mathbf{\Sigma}_0(\mathbf{v}\mathbf{r}^T)^T\right)+\operatorname{Tr}\left(\mathbf{\Sigma}_1^{-1}\mathbf{v}\mathbf{r}^T\mathbf{\Sigma}_0\right)+\operatorname{Tr}\left(\mathbf{\Sigma}_1^{-1}\mathbf{v}\mathbf{r}^T\mathbf{\Sigma}_0\mathbf{r}\mathbf{v}^T\right) \\
&\hspace{8cm}\scriptstyle{\operatorname{Tr}(ABC)=\operatorname{Tr}(CAB)=\ldots} \\
&= \sum_i \frac{\sigma_{0,i}^2}{\sigma_{1,i}^2}+\sum_i \frac{\sigma_{0,i}^2 r_i v_i}{\sigma_{1,i}^2}+\sum_i \frac{v_i r_i \sigma_{0,i}^2}{\sigma_{1,i}^2}+\operatorname{Tr}\left(\mathbf{v}^T\mathbf{\Sigma}_1^{-1}\mathbf{v}\mathbf{r}^T\mathbf{\Sigma}_0\mathbf{r}\right) \\
&= \sum_i \frac{\sigma_{0,i}^2+2\sigma_{0,i}^2 v_i r_i}{\sigma_{1,i}^2}+\sum_i r_i^2\sigma_i^2\cdot\sum_i \frac{v_i^2}{\sigma_i^2}.
\end{aligned}
$$

The last equation is easy to implement and only requires summing over the non-batch dimension. The difference of means can be derived very quickly with the same summing scheme:

$$
(\boldsymbol{\mu}_1-\boldsymbol{\mu}_0)^T\mathbf{\Sigma}_1^{-1}(\boldsymbol{\mu}_1-\boldsymbol{\mu}_0)=\sum_i \frac{(\boldsymbol{\mu}_1-\boldsymbol{\mu}_0)_i^2}{\sigma_{1,i}^2}.
$$

It remains the ratio of determinants, which we will simplify with the matrix determinant lemma giving

$$
\begin{aligned}
\log\left(\frac{\det\mathbf{\Sigma}_1}{\det\mathbf{A}\mathbf{\Sigma}_0\mathbf{A}^T}\right) &= \log\det\mathbf{\Sigma}_1-\log\det\left(\mathbf{A}\mathbf{\Sigma}_0\mathbf{A}^T\right) \\
&= \log\prod_i \sigma_{1,i}^2-\log\left(\det\mathbf{A}\cdot\det\mathbf{\Sigma}_0\cdot\det\mathbf{A}^T\right) \hspace{1cm}\scriptstyle{\det\mathbf{A}^T=\det\mathbf{A}} \\
&= 2\sum_i \log\sigma_{1,i}-\log\left((\det\mathbf{A})^2\prod_i \sigma_{0,i}^2\right) \hspace{1cm}\scriptstyle{\text{Matrix determinant lemma}} \\
&= 2\sum_i \log\sigma_{1,i}-\log\left(1+\mathbf{v}^T\mathbf{r}\right)^2-2\sum_i \log\sigma_{0,i} \\
&= 2\left(\sum_i\left(\log\sigma_{1,i}^2-\log\sigma_{0,i}^2\right)-\log(1+\sum_i v_i r_i)\right).
\end{aligned}
$$

Putting the above to formulas together finally yields

$$
\begin{aligned}
\operatorname{KL}(\mathcal{N}_0||\mathcal{N}_1)=\frac{1}{2}&\left(\sum_i \frac{\sigma_{0,i}^2+2\sigma_{0,i}^2 v_i r_i}{\sigma_{1,i}^2}+\sum_i r_i^2\sigma_i^2\cdot\sum_i \frac{v_i^2}{\sigma_i^2}\right. \hspace{2cm}(2)\\
&+\sum_i \frac{(\boldsymbol{\mu}_1-\boldsymbol{\mu}_0)_i^2}{\sigma_{1,i}^2}-k \\
&\left.+2\left(\sum_i\left(\log\sigma_{1,i}^2-\log\sigma_{0,i}^2\right)-\log(1+\sum_i v_i r_i)\right)\right).
\end{aligned}
$$

# 2  Supplementary to the experimental setup

## 2.1  Up-convolution

We used convolutional inference networks for the cart-pole and three-link arm task. While these networks help us overcome the problem of large input dimensionalities (i.e. $2\times 128\times 128$ pixel

images in the three-link arm task), we still have to generate full resolution images with the decoder network. For high-dimensional images generation fully connected neural networks are simply not an option. We thus decided to use up-convolutional networks, which were recently show to be powerful models for image generation [1, 2, 3].

To set-up these models we basically "mirror" the convolutional architecture used for the encoder. More specifically for each $5 \times 5$ convolution followed by $2 \times 2$ max-pooling step in the encoder network, we introduce a $2 \times 2$ up-sampling and $5 \times 5$ convolution step in the decoder network. The complete network architecture is given below. It is similar to the up-convolution networks used in Dosovitskiy et al. [2]. The upsampling strategy we use is simple "perforated" upsampling as described in [4].

## 2.2 Variational Autoencoder with slowness

Enforcing temporal slowness during learning has previously been found to be a good proxy for learning representations in reinforcement learning [5, 6] and representation learning from videos [7]. We also consider a VAE variant with a slowness term on the latent representation by enforcing similarity of the encodings of temporally close images. This can be achieved by augmenting the standard VAE objective $\mathcal{L}^{\text{bound}}$ with an additional KL divergence term on the latent posterior $Q_\phi$:

$$\mathcal{L}^{\text{slow}}(\mathbf{x}_t, \mathbf{x}_{t+1}) = \text{KL}(Q_\phi(\mathbf{z}_{t+1}|\mathbf{x}_{t+1})\|Q_\phi(\mathbf{z}_t|\mathbf{x}_t)). \tag{3}$$

Indeed there seems to be a slightly better coherence of similar states in the latent spaces, as e.g. depicted in Figure 4 in the main paper. Yet, our experiments show that a slowness term alone does not suffice to structure the latent space, such that locally linear predictions and control become feasible.

## 2.3 Evaluation criteria

For comparing the performance of all variants of E2C and the baselines, the following criteria are of importance:

- **Autoencoding**. Being able to reconstruct the given observations is the basic necessity for a model to work. The reconstruction cost drives a model to identify single states from its observations.

- **Decoding the next state**. For any planning to be possible at all, the decoder must be able to generate the correct images from transitions the dynamics model performed. If this is not the case, we know that the latent states of the encoding and the transition model do not coincide, thus preventing any planning.

- **Optimizing latent trajectory costs**. The action sequences for achieving a specified goal will be determined completely by locally linearized dynamics in the latent space. Therefore minimizing trajectory costs in latent space is, again, a necessity for successful control.

- **Optimizing real trajectory costs**. While the action sequence has been determined for the latent dynamics, the deciding criterion is whether this reflects the true state trajectory costs. Therefore carrying out the "dreamed" plans in reality is the optimality criterion for every model. To make the different models comparable, we use the same cost matrices for evaluation, which are not necessarily the same as for optimization.

We reflected these four criteria in the evaluation table in the paper. For the reconstruction of the current and next state we specified the mean log loss, which is in case of the Bernoulli distributions the cross entropy error function:

$$\log p(\mathbf{x}|\hat{\mathbf{x}}) = \frac{1}{N} \sum_{n=1}^{N} \sum_{i=0}^{n_x} x_{n,i} \log \hat{x}_{n,i} + (1 - x_{n,i}) \log(1 - \hat{x}_{n,i}). \tag{4}$$

For the costs a model imagines and truly achieves, we sample from different starting states and accumulate the distances in latent and true state space according to the SOC method.

## 2.4 The three-link robot arm

The robot arm we used in the last experiment in the main paper was simulated using dynamics generated by the MapleSim http://www.maplesoft.com/products/maplesim/ simulator wrapped in Python and visualized for producing inputs to E2C using PyGame. We simulated a fairly standard robot arm with three links. The length of the links were set to 2, 1.2 and 0.7 (units were set to meters). The masses of the corresponding links were all set to $10kg$.

## 2.5 Evaluating the true system model

To compare the efficacy of different models when combined with optimal control algorithms, we always reported the cost in latent space (as used by the optimal control algorithm) as well as the "real" trajectory cost. To compute this real cost, we evaluated the same cost function as in the latent space (quadratic costs on the deviation from a given goal state), but using the real system states during execution and different cost matrices for a fair comparison.

As an upper bound on the performance achievable for control by any of the models, we also computed the true system cost by applying iLQR/AICO to a model of the real system dynamics. We have this model available since all experiments were performed in simulation.

## 2.6 Neural Network training

### 2.6.1 Experimental Setup

All the datasets were created in advance as $\mathcal{D} = \{(\mathbf{x}_1, \mathbf{u}_1, \mathbf{x}_2), \ldots, (\mathbf{x}_{T-1}, \mathbf{u}_{T-1}, \mathbf{x}_T)\}$ for the training, validation and test split. While the E2C models were trained on $\mathcal{D}$, the ones that do not incorporate any transition information (i.e. AE, VAE) were trained on images $\mathcal{D}_{\text{images}} = \{\mathbf{x}_1, \ldots, \mathbf{x}_T\}$ extracted from the original dataset $\mathcal{D}$. The slowness VAE was trained on the pairs of images subset $\mathcal{D}_{\text{pairs}} = \{(\mathbf{x}_1, \mathbf{x}_2), \ldots, (\mathbf{x}_{T-1}, \mathbf{x}_T)\}$ and our E2C models on the full $\mathcal{D}$.

In order to learn dynamics predictions for the image-only autoencoders, we extracted the latent representations and combined them with the actions from $\mathcal{D}$ into $\mathcal{D}_{\text{dynamics}} = \{(\mathbf{z}_1, \mathbf{u}_1, \mathbf{z}_2), \ldots, (\mathbf{z}_{T-1}, \mathbf{u}_{T-1}, \mathbf{z}_T)\}$. On these low-dimensional representations we trained the dynamics MLPs, thus ensuring that all methods were trained on exactly the same data.

### 2.6.2 Implementation details

We used orthogonal weight initialization for every layer [8]. As described in the main paper, Adam [9] was used as the learning rule for all networks. We found both these techniques to be fundamentally important for stabilizing training and achieving good reconstructions for all methods. Both methods also clearly helped to cut the hyperparameter search needed for all methods to a minimum. In the process of training, we could make out three phases: the unfolding of the latent space, the overcoming of the trivial solution (the average image of the dataset) and the minimization of the latent KL term. The architectures used for our experiments were as follows (where ReLU stands for rectified linear units [10] and conv. for convolutions):

**Plane**

- Input: $40^2$ image dimensions, 2 action dimensions
- Latent Space dimensionality: 2
- Encoder: 150 ReLU - 150 ReLU - 150 ReLU - 4 Linear (2 for AE)
- Decoder: 200 ReLU - 200 ReLU - 1600 Linear (Sigmoid for AE)
- Dynamics: 100 ReLU - 100 ReLU + Output layer (except Global E2C)
    - AE, VAE, VAE with slowness, Non-linear E2C: 2 Linear
    - E2C: 8 Linear ($2 \cdot 2$ for $\mathbf{A}_t$, $2 \cdot 1$ for $\mathbf{B}_t$, 2 for $\mathbf{o}_t$), $\lambda = 0.25$
- Adam: $\alpha = 10^{-4}, \beta_2 = 0.1$
- Evaluation costs: $\mathbf{R}_z = 0.1 \cdot \mathbf{I}, \mathbf{R}_u = \mathbf{I}, \mathbf{R}_o = \mathbf{I}$

**Pendulum swing-up**

- Input: $2 \cdot 48^2$ image dimensions, 1 action dimension
- Latent Space dimensionality: 3
- Encoder: 800 ReLU - 800 ReLU - 6 Linear (3 for AE)
- Decoder: 800 ReLU - 800 ReLU - 4608 Linear (Sigmoid for AE)
- Dynamics: 100 ReLU - 100 ReLU + Output layer (except Global E2C)
    - AE, VAE, VAE with slowness, Non-linear E2C: 3 Linear
    - E2C: 12 Linear ($2 \cdot 3$ for $\mathbf{A}_t = (\mathbf{I} + \mathbf{v}_t \mathbf{r}_t^T)$, $3 \cdot 1$ for $\mathbf{B}_t$, 3 for $\mathbf{b}_t$), $\lambda = 0.25$
- Adam: $\alpha = 3 \cdot 10^{-4}, \beta_2 = 0.1$
- Evaluation costs: $\mathbf{R}_z = \mathbf{I}$, $\mathbf{R}_u = 0.1\mathbf{I}$, $\mathbf{R}_o = \mathbf{I}$

**Cart-Pole balancing**

- Input: $2 \cdot 80^2$ image dimensions, 1 action dimension
- Latent Space dimensionality: 8
- Encoder: $32 \times 5 \times 5$ ReLU - $32 \times 5 \times 5$ ReLU - $32 \times 5 \times 5$ ReLU - 512 ReLU - 512 ReLU
- Decoder: 512 ReLU - 512 ReLU - $2 \times 2$ up-sampling - $32 \times 5 \times 5$ ReLU - $2 \times 2$ up-sampling - $32 \times 5 \times 5$ ReLU - $2 \times 2$ up-sampling - $32 \times 5 \times 5$ conv. ReLU
- Dynamics: 200 ReLU - 200 ReLU + 32 Linear ($2 \cdot 8$ for $\mathbf{A}_t = (\mathbf{I} + \mathbf{v}_t \mathbf{r}_t^T)$, $8 \cdot 1$ for $\mathbf{B}_t$, 8 for $\mathbf{b}_t$), $\lambda = 1$
- Adam: $\alpha = 10^{-4}, \beta_2 = 0.1$
- Evaluation costs: $\mathbf{R}_z = \mathbf{I}$, $\mathbf{R}_u = \mathbf{I}$

**Three-link arm**

- Input: $2 \cdot 128^2$ image dimensions, 3 action dimensions
- Latent Space dimensionality: 8
- Encoder: $64 \times 5 \times 5$ conv. ReLU - $2 \times 2$ max-pooling - $32 \times 5 \times 5$ conv. ReLU - $2 \times 2$ max-pooling - $32 \times 5 \times 5$ conv. ReLU - $2 \times 2$ max-pooling - 512 ReLU - 512 ReLU
- Decoder: 512 ReLU - 512 ReLU - $2 \times 2$ up-sampling - $32 \times 5 \times 5$ ReLU - $2 \times 2$ up-sampling - $32 \times 5 \times 5$ ReLU - $2 \times 2$ up-sampling - $64 \times 5 \times 5$ conv. ReLU
- Dynamics: 200 ReLU - 200 ReLU + 48 Linear ($2 \cdot 8$ for $\mathbf{A}_t = (\mathbf{I} + \mathbf{v}_t \mathbf{r}_t^T)$, $8 \cdot 3$ for $\mathbf{B}_t$, 8 for $\mathbf{b}_t$), $\lambda = 1$
- Adam: $\alpha = 10^{-4}, \beta_2 = 0.1$
- Evaluation costs: $\mathbf{R}_z = \mathbf{I}$, $\mathbf{R}_u = 0.001\mathbf{I}$

Figure 1: Generated "dreamed" trajectories of different models for the plane task (from left to right). The opacity of the obstacles has been lowered in this depiction for better visibility of the agent.

# 3 Supplementary evaluations

## 3.1 Trajectories for plane and pendulum

To qualitatively measure the predictive accuracy, the starting state for a trajectory is encoded and the actions are applied on the latent representation. After each transition, the predicted latent position is decoded and visualized. In this manner, multi-step predictions can be generated for the planar system in Figure 1 and for the inverted pendulum in Figures 2 and 3.

Figure 2: Generated "dreamed" trajectories (from left to right) for *passive* dynamics: the pendulum starts with angle $\theta = -\frac{\pi}{2}$ without velocity. The models have to predict the dynamics, while no force is applied.

Figure 3: Dreamed trajectories (from left to right) for *controlled* dynamics: the pendulum starts with angle $\theta = \frac{\pi}{2}$ without velocity. For 6 timesteps, full force is applied to the right, followed by 4 timesteps of full force to the left.

## 3.2 Inverted pendulum latent space

Encoding the pendulum depictions into a 3-dimensional latent space allows for a visual comparison in Figure 4 .

Figure 4: Latent spaces of all baseline models and E2C variants for the inverted pendulum.

### 3.3 Trajectories for cart-pole and three-link arm

Finally – similar to the images in Section 3.1 – Figure 5 shows multi-step predictions for the cart-pole system. We depict important cases: (1) a long-term prediction with the cart-pole standing still (essentially the unstable fix-point of the underlying dynamics); (2) the cart-pole moving to the right, changing the direction of the poles angular velocity (middle column); (3) and the pole moving farthest to the right. The long-term predictions by the E2C model are all of high quality. Note that for the uncontrolled dynamics the predictions show a slight bias of the pole moving to the right (an effect that we consistently saw in trained models for the cart-pole). We attribute this problem to the fact that discretization errors in the image rendering process of the pole angle make it hard to predict small velocities accurately.

### 3.4 Exemplary trajectory taken for three-link arm task

Figure 6 shows a segment of a controlled trajectory for the three-link arm as executed by the E2C system. Note that, in contrast to other figures in this supplementary material, it does **not** show a long-term prediction but rather 10 steps of a trajectory (together with one-step-ahead predictions) that was taken by the E2C system when combined with model predictive control. For additional visualizations and controlled trajectories for all tasks we refer to the supplementary video.

### 3.5 Comparison of different models for cart-pole and robot arm

In Table 1 we compare our variety of models in terms of real trajectory cost and task success percentage on the cart-pole and the robot arm. All results are averaged over 30 different starting states with a fixed goal state.

The cart-pole always starts in the goal state (zero angle and zero velocity) with small additive Gaussian noise ($\sigma = 0.01$). Success is defined as preventing the pole from falling below an angle of $\pm 0.85$ rad. The three-link arm system begins in a random configuration and the goal is to to unroll all joints (e.g. make all angles zero) and stay $\epsilon$-close to that position.

The results show that only E2C and its non-linear variant can perform this task successfully, although there is still a large performance gap between the two. We conclude, that the error of linearizing non-linear dynamics after training the corresponding model grows to the point of no longer allowing accurate control for the system.

Figure 5: Dreamed trajectories (top to bottom) for uncontrolled (left column) and controlled (middle/right column) dynamics in the cart-pole system. The red image shows the initial configuration, which is encoded resulting in $\mathbf{z}_1$. The images in the right half of each column are then generated without additional input by following the dynamics in latent space. The left column depicts the uncontrolled case ($\mathbf{u} = 0$ for all steps). The middle column shows a controlled trajectory with torque $-20$ applied in each step and the right column a trajectory with torque 20 applied in each step. Prediction of the history image is omitted in these depictions.

Table 1: Comparison between trajectory costs of different approaches for the cart-pole and three-link task. The standard Autoencoder, Variational Autoencoder and Global E2C model are omitted from the table as they failed on this task (performance similar to VAE with slowness).

| Algorithm | True model | VAE + slownes | E2C no latent KL | Non-linear E2C | E2C |
|---|---|---|---|---|---|
| | | | **Cart-Pole balance** | | |
| **Traj. Cost** | $15.33 \pm 7.70$ | $49.12 \pm 16.94$ | $48.90 \pm 17.88$ | $31.96 \pm 13.26$ | $\mathbf{22.23 \pm 14.89}$ |
| **Success** % | 100 % | 0 % | 0 % | 63 % | **93** % |
| | | | **Three-link arm** | | |
| **Traj. Cost** | 59.46 | $1275.53 \pm 864.66$ | $1246.69 \pm 262.6$ | $460.40 \pm 82.18$ | $\mathbf{90.23 \pm 47.38}$ |
| **Success** % | 100 % | 0 % | 0 % | 40 % | **90** % |

Table 2: Comparison between AICO and iLQR based on the "real" cost for controlling the cart-pole and three-link robot arm using convolutional networks.

| Method | iLQR | AICO |
|---|---|---|
| **Cart-Pole** | | |
| E2C | $14.56 \pm 4.12$ | $12.56 \pm 2.47$ |
| True model | $7.45 \pm 1.22$ | $7.03 \pm 1.07$ |
| **Three-Link Robot Arm** | | |
| E2C | $93.78 \pm 32.98$ | $92.99 \pm 20.12$ |
| True model | $53.59 \pm 9.74$ | $56.34 \pm 10.82$ |

## 3.6 Comparison of trajectory optimizers for cart-pole and robot arm

To compare how well AICO deals with the covariance matrices estimated in latent space we performed an additional experiment on the cart-pole and three-link robot arm task comparing it to iLQR. We performed model predictive control using the locally linear E2C model starting in 10 different start states each. The remaining settings are as given in Section 3.5.

As reported in Table 2, both methods performed about the same for these tasks, indicating that the covariance matrices estimated by our model do not "hurt" planning, but considering them does not improve performance either.

Figure 6: Frames extracted from a trajectory (top to bottom) as executed by the Embed to Control system. The left column shows the real images corresponding to transitions taken in the MDP. Middle and right column show the prediction of history image and current image based on the previous two images.