[Reviews · NeurIPS 2015]

Submitted by Assigned_Reviewer_1

This paper proposes a combination of deep learning and stochastic optimal control for control of non-linear dynamical systems from image inputs. The system learns a latent space representation of the true system state based on images of the state using deep variational auto encoders. Additionally, a locally-linear dynamics model is learned on the latent space which is used to generate state space trajectories by applying stochastic optimal control techniques directly on the latent space. The system is tested on four control tasks and the performance is compared with multiple baselines showing good performance.

Quality: --------- The paper is logical and sound. The use of (deep) variational encoders to learn the latent space representation mirrors previous work. Enforcing local-linearity on the transition dynamics coupled with the constraint between the transition and inference distributions is crucial for good performance. It would be interesting to see if the authors can model longer term dependencies using RNNs and/or LSTMs. The experiments highlight the strength of the approach, especially the fact that the learned representations are conducive for control. More experiments on higher dimensional (state/control dimensionality) tasks would help. The authors could compare their learned latent representations with those learned by Predictive State Representation based methods.

Clarity: --------- The paper is very well written and is easily understandable. The fact that the authors give details on the structure of their networks and various hyper-parameters in the supplementary materials is appreciated. In Sec 2.2, line 103, correct c_T(x_t,u_t) = c(x_T, u_T).

Originality: ------------- The main novelty of the paper lies in controlling the system directly on the latent space and the restrictions on the latent transition dynamics. This makes the method more stable and improves performance significantly as compared to prior approaches.

Significance: --------------- The paper advances the state of the art in pixel based control of non-linear dynamical systems and is very relevant to the deep learning, RL and representation learning community.

Related citation to add: Boots et al., Closing the Learning-Planning Loop with Predictive State Representations, RSS 2010 -> The authors learn a model of the environment from camera data and plan on the learned model

I vote for a clear accept.
Summary: This paper presents a system for controlling non-linear dynamical systems from raw pixel data by learning a latent space representation with an added dynamics model and applying stochastic optimal control techniques directly on the latent space. The paper is very well written, the idea is novel with strong experimental results. I vote for a clear accept.

Submitted by Assigned_Reviewer_2

This paper presents a framework for learning and controlling non-linear dynamical systems based on high-dimensional observations (for example raw pixels). The solution uses variational auto-encoders for dimensionality-reduction and an optimal control formulation in the low-dimensional latent space to compute controls.

This is an interesting model-based approach for computing continuous controls directly from image pixels with (what it seems like) promising results. I like the use of an embedded low-dimensional dynamical model and how this has been implemented to reach a (fairly) probabilistic formulation. The model is well explained and the authors have also implemented and compared with many different related model structures which strengthen the results.

The result section is (as far as I can see) lacking quantitative results on the quality of the controller. For example, by doing several MC runs, what is the rate for completing the task successfully (possibly as a function of number of frames)? With the present material in the result section it is difficult to judge the quality of the controller. Further, in the related recent work section, you should also include [1], which also as a similar example with the inverted pendulum as you have.

Overall, I think this a good paper that should be accepted for the conference.

Minors:

- Section 2.1: I don't understand the connection between s_t and x_t. Could you please explain that more carefully?

- Shouldn't all your noise system terms, xi in (1) and omega in (4), have an index t as well?

- Row 158. Should the \omega really be included in the mean? Its mean is zero and the covariance is included in C_t already.

[1] H. van Hoof, J. Peters, and G. Neumann. Learning of non-parametric control policies with high-dimensional state features. In AISTATS, 2015.
Summary: This is an interesting model-based approach for computing continuous controls directly from image pixels with (what it seems like) promising results.

Submitted by Assigned_Reviewer_3

The authors propose a method for model-based control of nonlinear dynamical systems from visual observations. The approach involves learning a low-dimensional latent representation of the system by predicting future image observations using a variant of the variational autoencoder with a temporal prediction component. Furthermore, the model directly outputs a "linearization" of itself, which can be used with efficient MPC methods to actually determine a good sequence of controls.

I think the idea behind this paper is quite nice, and it's very interesting to see that the model which outputs linearizations appears to actually perform worse at prediction but *better* at control.

The technical approach is quite logical and seems to me to be sound.

The biggest weakenesses of the paper are novelty and evaluation.

Novelty is an issue because a number of prior methods have been proposed that attempt to do nearly the same thing. Some of these are covered in Sec 4, with [24] and [21] seemingly the most relevant. However, there is also substantial work on control from images using kernel-based representations (see, e.g., Non-Parametric Policy Learning for High-Dimensional State Representations, though there are many others), with results shown on similar kinds of tasks. In light of the previous work in this area, it seems that the principal contributions are the use of VAE and the linearization trick. These are both nice, but one is an application of an existing method, and the other is quite simple.

The simplicity of this trick would be less of an issue if the authors showed conclusively that it makes a big difference. However, the method is only rigorously evaluated on toy problems (2D navigation and pendulum). The more complex tasks do not have a rigorous set of comparisons, and it's unclear how much the proposed components really make a difference here. Since there is little theoretical support for the usefulness of these improvements, I believe that stronger theoretical results are really necessary to make a strong case for the effectiveness of these tricks, even though I myself find them intuitively very appealing.
Summary: An interesting method for control from images that uses deep networks to embed the images in a space with certain local linearity properties that make it straightforward to apply MPC, which is a nice and quite promising idea.

Author Feedback
Author rebuttal: We thank all reviewers for their thorough, positive reviews and insightful comments.

*All reviewers:
[novelty as an issue]
We agree upon a connection between our work and [21,24]. [21] is equivalent to the AE included in our comparison (Table 1). [24] is similar to the Non-Linear E2C model. Important differences are:
1) We use a variational formulation which gives rise to a Gaussian latent space; allows for exact propagation of latent Gaussian dynamics
2) We enforce agreement between the Markov chain in the latent and observed space (via the KL term in Eq. 11)
3) We enforce an explicit local linearization, valid *for all control magnitudes*
These result in (we believe) the first learned latent state space model that directly complies with the iLQG requirements. Our experiments show that robust control is only possible when all 3 components are used (also see below).

[reg. evaluation metrics/more thorough evaluation]
We agree that we should have included more interpretable evaluation metrics. We performed additional experiments on all domains computing success percentages (defined as reaching the goal and staying epsilon-close) from 30 start states. The results for planar and swing-up: Only the E2C variants reached the goal. Non-Linear E2C in 96%/46% (planar/swing-up) of the cases and E2C in 100%/90% respectively. Removing the KL constraint from E2C results in failure again. For cart-pole and robot-arm: No non-E2C variant resulted in stable control. Non-Linear-E2C reached a 63%/40% success ratio while E2C reached 93%/90% respectively. An updated results table can be accessed here: https://goo.gl/g1HxmN it will be included in the final version.

[reg. comparisons to previous work/selected baselines are very weak]
While a comparison to more approaches would be interesting, the goal of our paper is to enable model based control from raw images. We respectfully disagree that for this setting our baselines are weak. VAEs arguably constitute the state of the art for generative models [7,26], the only other system for latent model based control uses AEs [24]. We thus focused on AEs/VAEs as baselines (including a novel VAE with slowness).
Moreover, a fair comparison to DQN [22] is not easily possible as our domains require continuous (multi-dimensional) actions. DQN is currently restricted to discrete actions. Extending it to cont. actions is important but out of scope for this paper. Preliminary experiments with neural Q-learning with discretized actions failed on our, small, fixed sample sets.

[reg. kernel-based RL from raw images]
Thank you for the reference to van Hoof et al.; which we will include. We believe their work to be orthogonal to ours as they consider model-free policy learning; only implicitly dealing with system dynamics/embeddings. Applying their approach to our domains will require additional engineering since: 1) they only deal with 20x20 pixel images; 2) a squared exponential kernel on pixels might not be a reasonable metric for complicated images.

*Meta_Reviewer:
Please see "All reviewers" regarding additional experiments and used baselines. An objective not based on reconstruction would be great but we are not aware of one applicable in our setting. We want to clarify that our method is not restricted to binary images. As for other VAEs the output distribution can easily be changed (e.g. to be Gaussian)[6,7].

[where do costs c(.) and/or J(.)) appear in Eq. 12]
This seems to be a misunderstanding regarding our objective. The control costs are *not* included in Eq. 12. since they are unnecessary for inferring the latent space (due to duality of control and model learning); they are only subsequently used to perform control (cf. Section 3.1). We are sorry for the confusion and will stress this in the final version.

[reg. network architecture/overcomplete representations]
Great question. We performed experiments with different network architectures prior to those presented in the paper. In brief: we did not find large latent representations to improve control performance for any algorithm (including AEs), but they make optimal control unbearably slow.

*Reviewer_1+Reviewer_9:
Thank you for your positive comments, error spotting, and the pointer to PSRs. We'll discuss them in the final version. For more experiments please see "All reviewers" above.

[combining E2C with RNNs to model longer term dependencies]
We fully agree and believe that this is a logical next step.

[minor comments]
Errors regarding terminal costs, \omega (which should not contribute to the mean in Eq. 8) and \xi will be fixed. s_t is the true system state (e.g. angle/velocity) it is partially observable through the rendered image x_t. We will clarify the terminology.

*Reviewer_3:
Thank you for very helpful comments. Please see "All reviewers" above regarding your main criticism. We agree and will highlight differences to prior art and importance of E2C components more clearly.